# Antiviral Evaluation of New Synthetic Bioconjugates Based on GA-Hecate: A New Class of Antivirals Targeting Different Steps of Zika Virus Replication

**DOI:** 10.3390/molecules28134884

**Published:** 2023-06-21

**Authors:** Paulo Ricardo da Silva Sanches, Ricardo Sanchez-Velazquez, Mariana Nogueira Batista, Bruno Moreira Carneiro, Cintia Bittar, Giuditta De Lorenzo, Paula Rahal, Arvind H. Patel, Eduardo Maffud Cilli

**Affiliations:** 1School of Pharmaceutical Science, São Paulo State University, Araraquara 14800-903, SP, Brazil; 2MRC—University of Glasgow Centre for Virus Research, University of Glasgow, Glasgow G12 8QQ, UK; ricardo.sanchez@biontech.de (R.S.-V.); giuditta.delorenzo@glasgow.ac.uk (G.D.L.); arvind.patel@glasgow.ac.uk (A.H.P.); 3Institute of Chemistry, São Paulo State University, Araraquara 14800-900, SP, Brazil; 4Laboratory of Virology and Infectious Diseases, The Rockefeller University, New York, NY 10065, USA; mnogueirab@rockefeller.edu (M.N.B.);; 5School of Health Science, Federal University of Rondonópolis, Rondonópolis 78736-900, MT, Brazil; bruno@ufr.edu.br; 6Institute of Bioscience, Humanities and Exact Science, São Paulo State University, São José do Rio Preto 15054-000, SP, Brazil; p.rahal@unesp.br

**Keywords:** Zika virus, gallic acid, peptides, Hecate, bioconjugates, GA-Hecate, GA-metabolites

## Abstract

Re-emerging arboviruses represent a serious health problem due to their rapid vector-mediated spread, mainly in urban tropical areas. The 2013–2015 Zika virus (ZIKV) outbreak in South and Central America has been associated with cases of microcephaly in newborns and Guillain–Barret syndrome. We previously showed that the conjugate gallic acid—Hecate (GA-FALALKALKKALKKLKKALKKAL-CONH_2_)—is an efficient inhibitor of the hepatitis C virus. Here, we show that the Hecate peptide is degraded in human blood serum into three major metabolites. These metabolites conjugated with gallic acid were synthesized and their effect on ZIKV replication in cultured cells was evaluated. The GA-metabolite 5 (GA-FALALKALKKALKKL-COOH) was the most efficient in inhibiting two ZIKV strains of African and Asian lineage at the stage of both virus entry (virucidal and protective) and replication (post-entry). We also demonstrate that GA-metabolite 5 does not affect cell growth after 7 days of continuous treatment. Thus, this study identifies a new synthetic antiviral compound targeting different steps of ZIKV replication in vitro and with the potential for broad reactivity against other flaviviruses. Our work highlights a promising strategy for the development of new antivirals based on peptide metabolism and bioconjugation.

## 1. Introduction

Zika virus (ZIKV) is an arbovirus belonging to the *Flaviviridae* family, members of which include dengue virus (DENV), yellow fever virus (YFV), West Nile virus (WNV), Japanese encephalitis virus (JEV), and tick-borne encephalitis virus (TBEV) [1,2,3,4]. They are all enveloped viruses carrying a positive single-stranded RNA genome [5,6,7].

ZIKV is widespread throughout the tropics, with local variations influenced by rainfall, temperature, relative humidity, and unplanned rapid urbanization [8,9,10,11]. Transmission occurs mainly via the bite of an infected mosquito from the *Aedes* genus, such as *Aedes aegypti* and *Aedes albopictus*. Other routes of infection include blood transfusion and sexual intercourse [12,13,14], but the most devastating complications follow the vertical transmission of the virus from a pregnant human to the foetus, causing serious birth defects known as ZIKV congenital syndrome [15]. ZIKV infection is also associated with Guillain–Barre syndrome (GBS), a post-infectious autoimmune disorder identified by bilateral flaccid limb weakness attributable to peripheral nerve damage [16,17]. Since 2014, serologic and molecular testing has implicated ZIKV in many cases of illness in Brazil [18,19]. In 2015, autochthonous transmission in Brazil was confirmed and, since then, ZIKV has spread extensively throughout South and Central America, the Caribbean, and the USA [11,20,21]. In 2019, a total of 87 countries and territories reported autochthonous cases of infection. According to the WHO, as of June 2020, more than 1.3 million cases of ZIKA virus infection have been reported worldwide.

Flavivirus virions share common characteristics such as a spherical particle of 40–60 nm in diameter, enveloped by a host-derived lipid bilayer containing the viral glycoproteins called the envelope (E) and membrane (M), which in immature particles is present as precursor membrane protein (prM). These glycoproteins [22,23,24] play a role in virus tropism, host cell entry, host immune response, and other important processes [22]. 

The viral genome consists of 5′ and 3′ untranslated regions (UTRs), flanking a single open reading frame (ORF) encoding a polyprotein precursor of about 3430 amino acids that is post-translationally processed into three structural proteins (C, prM, and E) and seven nonstructural proteins (NS1, NS2A, NS2B, NS3, NS4A, NS4B, and NS5) [22,23,24,25,26,27,28,29,30,31,32].

Flavivirus enter host cells through receptor-mediated endocytosis [28,29,30]. Once the virus is in the endosome, the acidic environment triggers the irreversible trimerization of the E protein, resulting in the fusion of viral and endosome membranes [32]. The RNA genome is then released into the cytoplasm, where it is translated into a polyprotein which is then processed by the host and viral proteases into mature proteins that participate in viral RNA synthesis, particle assembly, maturation, and release [32]. Currently, there are no approved vaccines or drugs to prevent or treat ZIKV infection, and as such, their development is of topical interest. 

Antiviral drugs that block virus infection can be discriminated based on the step of the viral cycle that they affect, such as virus attachment, entry, virus uncoating and genome release, viral protein synthesis, virus maturation, and release [33,34,35,36,37]. As viruses depend on cell machinery for their replication, an effective antiviral agent must prevent virus multiplication without disturbing cellular functions. 

Peptides with antiviral properties are advantageous because they can be rationally designed and exhibit highly diverse structures and broad biological activities [38,39,40,41]. Several synthetic antiviral peptides have been reported such as the peptide C5A that prevents hepatitis C virus (HCV) infection by inactivating extra and intracellular infectious particles [42]. Peptides derived from E protein of JEV were shown to reduce histopathological damage in the brain and testis of ZIKV-infected type I and II interferon receptor-deficient mice [43]. The Hecate peptide, an amphipathic α-helical peptide, reduced herpes simplex virus type1 plaque formation and inhibited virus-induced cell fusion and virus spread without having toxic effects on eukaryotic cells [44]. Additionally, synthetic peptides have shown high potential against SARS-CoV-2 targeting the PL^pro^ non-structural protein in vitro [45]. In previous studies, we showed that the association of gallic acid (GA) at the N-terminus of the Hecate lytic peptide (H_3_^+^N-FALALKALKKALKKLKKALKKAL-CONH_2_) promoted changes in its biological activity, decreasing its toxicity in non-tumour keratinocyte cells [46]. We thus classified gallic acid-associated Hecate (GA-Hecate/GA-FALALKALKKALKKLKKALKKAL-CONH_2_) as a new class of antiviral against HCV (*Flaviviridae* family, *Hepacivirus* genus), being the most efficient and non-toxic Hecate peptide derivative [33]. GA-Hecate targeted all major steps in the HCV infectious cycle, inhibiting the infection of cells by the viral genotypes 2a and 3a by 50% to 99%. Furthermore, the peptide interacted with host lipid droplets and intercalated with RNA. 

In this study, we evaluate the serum stability of the Hecate peptide, identify the prevalent metabolites, and evaluate their synthetic gallic-acid-conjugated derivatives for their potential to inhibit different steps of ZIKV replication.

## 2. Results

### 2.1. Assessment of Hecate Stability in Human Serum

The GA-Hecate peptide has shown promising efficacy in antiviral assays, but its stability and target selectivity are yet to be determined. An understanding of these properties is crucial for the future progression of the peptide for therapeutic end use. To evaluate its stability, the Hecate peptide was incubated in freshly prepared human blood serum at 37 °C for 20 min. Mass spectrometry was then employed to identify breakdown products (metabolites) (Appendix A). As shown in Figure 1, using the “molecular weight/charge” (*m*/*z*) relationship, N- and C-terminal fragments of different lengths—arbitrarily named metabolite 5, 6, and 7 (N-terminal or -C-terminal)—were identified and their respective cleavage sites deduced (Figure 1 and Table 1). 

### 2.2. Peptide Synthesis

The Hecate N-terminal fragments found in the serum stability experiment were selected to conjugate with gallic acid, like that in the GA-Hecate structure, where GA is coupled at the N-terminal region. Bioconjugates were synthesized using solid-phase methods and, after purification, their purity and identities were determined by high-performance liquid chromatography and mass spectrometry, respectively (see Appendix A). The chemical properties of the synthesized compounds are shown in Table 2. GA-metabolite 6 was not water-soluble and therefore was not considered for further biological evaluation. 

### 2.3. The Cytotoxicity Profile of GA-Peptides

To understand the effect of GA-Hecate and N-terminal-GA-metabolites on host cells, we tested them at a range of concentrations (80–0.62 µM) on Vero cells. Considering the arbitrary threshold of 80% cell viability, GA-Hecate, and GA-metabolite 5 reached their maximum non-toxic concentration at 40 µM (Figure 2). GA-metabolite 7 did not show any cytotoxic effect at all tested concentrations. 

### 2.4. Antiviral Activity

The activity of GA-peptides in different replication steps of the ZIKV strain PE243 (Asian lineage isolated in Recife, Brazil, during the outbreak in 2015—ZIKV^BR^)^51^ and the ZIKV strain MP1751 (African lineage isolated in Uganda in 1962—ZIKV^AF^)^50^ was evaluated [47,48].

#### 2.4.1. Post-Entry Assessment of GA-Hecate and GA-Metabolites 

Vero cells were infected with the ZIKV strain PE243 at an MOI of 0.1 for 1 h after which they were washed with PBS, and then a fresh medium containing different concentrations of GA-Hecate, GA-metabolite 5, or GA-metabolite 7 was added. At 72 h post-infection (hpi), cells were lysed, and the level of viral E protein was determined by sandwich ELISA. The amount of E protein in this assay correlates with that of the viral replication, thus allowing us to quantitatively assess the inhibitory potential of the GA-peptides and determine the concentration at which 50% of virus replication is inhibited (effective concentration—IC_50_) [49].

Both GA-Hecate and GA-metabolite 5 inhibited ZIKV strain PE243 replication in a dose-dependent fashion (Figure 3b,c), while GA-metabolite 7 was ineffective (Figure 3d). GA-Hecate inhibited viral replication by 60%, 40% and 20% at the non-toxic concentrations of 40, 20 and 10 µM, respectively (Figure 3b). GA-metabolite 5, on the other hand, was only partially efficient, reducing virus replication by 40% at 40 µM concentration (Figure 3c). We next evaluated the effect of GA-Hecate and GA-metabolite 5 combined (co-treatment) under the same conditions. No enhancement in the inhibition of infection was observed upon the co-treatment of infected cells with GA-Hecate and GA-metabolite 5, and the inhibitory levels were similar to those conferred by GA-Hecate alone (Figure 3e). The relative IC_50_ (50% inhibitory concentration) and CC_50_ (50% cytotoxicity concentration) derived from the Figure 3 data are shown in Table 3.

The effect of GA-Hecate and GA-metabolite 5 on the ZIKV strain MP1751 was also evaluated. Vero cells were infected and GA-Hecate or GA-metabolite 5 was added to the cells at the concentration of 20 or 40 µM. Following incubation at 37 °C for 48 or 72 hpi, the levels of viral replication were determined as per the sandwich ELISA described above. The 48 hpi time point was employed as ZIKV strain MP1751 has a faster replication rate than ZIKV strain PE243 in cultured cells. As shown in Figure 4, treatment with GA-Hecate at 40 µM and 20 µM for 72 h decreased the virus replication rate by 53% and 26%, respectively, while GA-metabolite 5 was ineffective (Figure 4c—72 h). GA-Hecate conferred similar levels of viral replication inhibition at 48 h post-infection (Figure 4b). Thus, GA-Hecate was equally effective against both ZIKV strains PE243 and ZIKV strain MP1751. In contrast, while GA-metabolite 5 displayed an inhibitory effect on ZIKV strain PE243, it failed to block the replication of ZIKV strain MP1751 at 72 hpi (Figure 4c), although at 40 µM concentration, a 57% reduction in virus replication was observed at 48 hpi (Figure 4b). These results indicate that GA-metabolite 5 affects ZIKV strain MP1751 in the first 48 h, but the virus quickly recovers thereafter. The ZIKV strain MP1751 is highly efficient at infection and replication and therefore is likely to overcome the inhibitory effect of GA-metabolite 5 upon longer incubation [51]. Even though the GA-metabolite 5 presents higher uptake efficiency, the results show that GA-Hecate is the most efficient compound in the post-entry assays in both ZIKV strains. 

#### 2.4.2. Virucidal Effects of GA-Hecate and GA-Metabolite 5

Targeting virus particles is an important prophylactic strategy to prevent virus infection. To test whether GA-Hecate and GA-metabolite 5 can directly inactivate virus, they were first incubated at different concentrations, either individually or in combination, with ZIKV strain PE243 at an equivalent of 0.1 MOI for 1 h at room temperature. The mixture was used to infect Vero cells for 1 h at 37 °C. After washing with PBS, the cells were incubated for 72 h when the virus replication levels were determined using the micro-neutralization assay. As shown in Figure 5b, GA-Hecate at 40 µM inhibited the replication of ZIKV strain PE243 by approximately 47%, but no inhibition was seen at lower concentrations. In contrast, under these conditions, the GA-metabolite 5 inhibited ZIKV strain PE243 infectivity by 74.6% at both 40 and 20 µM, and by 45.9% at 10 µM concentration (Figure 5b). The co-treatment showed a decrease in inhibition compared with GA-metabolite 5 at 20 and 10 µM. As GA-Hecate did not show virucidal activity at these concentrations, the molecule could be avoiding the GA-metabolite 5 to interact with the virus particle and block the virucidal effects. 

We next tested the virucidal effect of GA-Hecate and GA-metabolite 5 at concentrations of 20 µM and 40 µM on ZIKV strain MP1751 as described above. Again, we found a modest effect of GA-Hecate at 40 µM on ZIKV strain MP1751 replication. Surprisingly, however, GA-metabolite 5 was even more efficacious under these conditions against ZIKV strain MP1751, displaying almost complete (98.5%) inhibition at both 40 µM and 20 µM concentrations (Figure 5c). Given the latter result, the combination was not tested.

The progression of virus infection over time following the pre-treatment of virus particles with the GA-peptides was also evaluated using fluorescence microscopy. ZIKV strain MP1751 was pre-incubated (virucidal treatment) with 20 or 40 µM of GA-Hecate or GA-metabolite 5, and the treated virus was used to infect Vero cells for 1 h. After a PBS wash, the cells were incubated for 24, 48, and 72 h, and then fixed with ice-cold methanol. The viral E protein was detected via an indirect immunofluorescence assay (IFA). As expected, the progression of infection over time was readily observed in cells infected with untreated virus (Figure 6a). In cells infected with the virus treated with 40 µM GA-Hecate, the progression infection was low with only a few weakly E-expressing cells visible at 48 hpi (Figure 6a), although the course of infection in cells treated with the 20 µM of the peptide was like the untreated control (Figure 6b). In keeping with the data shown in Figure 5b, there were no signs of infection in cells infected with the virus treated with GA-metabolite 5 at both concentrations tested, and at both time points (Figure 6a,b). Together, these results indicate that both GA-Hecate and GA-metabolite 5 also have an antiviral effect on virus particles, with the latter being extremely potent.

We next assessed the effect of both GA-peptides using a focus-forming assay, which quantitatively showed the concentration of virus released in the supernatant after virucidal experiments. Vero cells were infected at 0.1 MOI of ZIKV strain MP1751, which had been pre-treated with 20 or 40 µM of GA-Hecate or GA-metabolite 5 for 1 h. At 72 hpi, the amount of infectious progeny virus released into the cell medium was quantitated using a focus-forming assay, as described using the Spaerman–Karber method [52]. Consistent with IFA, no infectious particles were detected at both 48 and 72 hpi for the virus treated with GA-metabolite 5 at 40 µM, and a very small number of infectious units were observed after treatment with 20 µM of the peptide (Table 4). In contrast, the pre-treatment of the virus with 40 µM GA-Hecate was only effective at 48 hpi when a small amount of infectious virus was found. This number increased upon the further incubation of infected cells to 72 hpi. 

#### 2.4.3. Protective Effect of GA-Hecate and GA-Metabolite 5

During the virucidal experiments, the GA-peptide-treated virus was incubated with the cells, during which it could interact with the cell’s surface and contribute to the high levels of inhibition seen above. To better evaluate the protective effects of peptides, Vero cells were first incubated with different concentrations of the peptides for 1 h, washed with PBS, infected with strain ZIKV strain PE243 or ZIKV strain MP1751 at an MOI of 0.1 for 1 h, and then washed again, followed by further incubation at 37 °C for 72 h. Then, cells were lysed, and the E protein levels were determined using the sandwich ELISA described above. The results show that GA-Hecate inhibited ZIKV strain PE243 replication by 43.9% and GA-metabolite 5 by 16.8% at 40 µM (Figure 7b). No inhibition was seen with either peptide when used at a lower concentration of 20 and 10 µM. Interestingly, the co-treatment of cells with GA-Hecate and GA-metabolite 5 at 40 µM and 20 µM, respectively, resulted in 92.4% and 20.8% inhibition of ZIKV strain PE243 (Figure 7b). GA-Hecate only slightly reduced the replication of ZIKV strain MP1751 which was statistically insignificant (Figure 7c). In contrast, GA-metabolite 5 significantly inhibited (by 86.7%) ZIKV strain MP1751 at 40 µM both on its own and in combination with GA-Hecate. This effect was not reproduced when the peptide(s) were used at a lower concentration of 20 µM (Figure 7c). 

#### 2.4.4. Effect of GA-Hecate and GA-Metabolite 5 on Virus Assembly and Release

To assess the effects of peptides on virus assembly and release, Vero cells were first infected with ZIKV strain PE243 or MP1752 at 0.1 MOI. After 1 h, cells were washed with PBS and fresh medium containing 40 or 20 µM of GA-Hecate or GA-metabolite 5 was added. At 48 hpi, the medium was removed, cells were washed three times with PBS and then they were incubated further at 37 °C in 2% FBS-supplemented culture medium for an additional 24 h. The supernatant of these cells was collected and used as an inoculum to infect naive Vero cells for 1 h. After being washed with PBS, cells were incubated for an additional 72 hpi and lysed, and the amount of viral protein E was measured using the sandwich ELISA. As shown in Figure 8a,b, no reduction in infectivity relative to the control was seen, indicating that the peptides do not affect virus assembly and release.

### 2.5. GA-Hecate and GA-Metabolite Cell Toxicity during Constant Treatment

We next assessed the effect, if any, of our peptides on cell growth, which in turn could affect virus replication. Vero cells in culture flasks were propagated in the presence or absence of 20 µM each of GA-peptides for 3 and 7 days. At this time point, viable cells were counted following incubation with trypsin using the automated cell counter Scepter^TM^ (Millipore), and cell shape was monitored using phase-contrast microscopy. Then, 7 × 10^3^ cells were seeded per well of a 96-well plate, incubated for 24 h, and cell viability was measured using the WST-1 viability reagent (Figure 9c,d). We found that the number of cells was similar in all treatments; however, cell shape was notably different in cells treated with GA-Hecate (Figure 10a,b). This was consistent with the lower cell viability found for GA-Hecate-treated cells (65% of the control, Figure 9c). In contrast, GA-metabolite 5 did not have any effect on cell viability or cell shape (Figure 9a–d). 

Based on our results, we propose a possible mode of action of GA-metabolite 5 (Figure 10). The virus replication rate decreased as a result of mutual factors involved in viral particles disrupting and blocking cellular receptors related to virus infection in the extracellular space, and bioconjugate/ssRNA(+) and bioconjugate/viral enzymes interacting with the synthesis of new virus particles in the intracellular space. Evaluation of the entry steps (virucidal and protective) suggested that virus infection was blocked by GA-metabolite 5, substantially decreasing the virus replication rate. Since laboratory virus strains may develop mutations that could affect virus receptor binding, we must be careful and evaluate these protective effects in recently isolated clinical strains. However, virucidal and protective results based on ZIKV strain PE243 and ZIKV strain MP1751 revealed a promising compound with prophylactic effects. Complementary evaluation of the post-entry steps (cytoplasm replication) showed that GA-metabolite 5 was able to reduce virus replication rate in a time/concentration-dependent manner. Even though GA-Hecate presented more efficient inhibition in the post-entry steps than GA-metabolite 5, this compound altered cell metabolism/growth and, consequently, affected virus replication. 

## 3. Materials and Methods

Serum stability. A total of 100 µL of a solution containing 1 mg/mL of Hecate peptide was incubated at 37 °C with 100 µL of fresh human serum for 20 min. Then, 200 µL of 96% ethanol solution in water was added to the mixture, which was centrifuged, and the supernatant was analysed by mass spectrometry as described below. Water was used as a negative control for samples without serum [53,54] (Appendix A).

Peptide synthesis. Peptide synthesis was performed via the automated solid-phase process using the standard Fmoc (9-fluorenylmethyloxycarbonyl) protocol in a TRIBUTE-UV (protein) synthesizer on a Rink-MBHA resin (Hecate, GA-Hecate) and Wang resin (GA-metabolites). Compounds were purified by high-performance liquid chromatography (HPLC) and the identity was confirmed by electrospray mass spectrometry [33,46] (Appendix A)

Mass spectrometry. Compound solutions were analysed by mass spectrometry using a Bruker spectrometer in electrospray positive mode with direct injection. Spectra were analysed with the help of Spectra Analysis Software using the relationship between molecular weight (m) and charge (z) (*m*/*z*). Fragments of Hecate were identified, and the amino acid sequence was determined by *m*/*z* comparison. 

Cells and virus. The African green monkey epithelial kidney cell line Vero (ATCC CCL81) was cultured in Dulbecco’s modified Eagle’s medium (DMEM—Gibco—Life Technologies, Grand Island, NY, USA) supplemented with 10% fetal bovine serum, 100 IU/mL penicillin, 100 µg/mL streptomycin, and 1% (*v*/*v*) non-essential amino acids (Gibco—Life Technologies, USA) and maintained at 37 °C in a humidified 5% CO_2_ incubator [33]. ZIKV strain PE243 was described in the paper of Donald et al. and ZIKV strain MP1751 (005V-02871) was kindly supplied by Public Health England (accession number KY288905.1)

The cytotoxicity profile of GA-Hecate and GA-metabolites. Peptide cytotoxicity profiles were measured using the WST-1 (4-[3-(4-iodophenyl)-2-(4-nitrophenyl)-2H-5-tetrazolio]-3-benzene disulfonate) cell proliferation reagent. A total of 24 h before treatment, 7 × 10^3^ Vero cells/well were cultured with supplemented DMEM in a 96-multi-well plate and incubated at 37 °C in a humidified 5% CO_2_ incubator. The next day, the peptide-containing medium was added at the following final concentrations: 80; 40; 20; 10; 5; 2.5; 1.25, and 0.625 µM. After 72 h of incubation, peptide medium was removed, and WST-1 at a dilution of 1:10 was added to the wells and incubated for 30 min. Absorbance was measured at 450 nm, using a plate reader spectrophotometer.

Post-entry assessment of GA-Hecate and GA-metabolites on Zika Virus. Vero cells (7 × 10^3^ cells/well) were infected for 1 h with the ZIKV strain PE243 (MOI 0.1) and then treated with different concentrations of peptide solution (ranging from 80 µM to 0.62 µM) in a 2% FBS culture medium. 

Sandwich ELISA to assess ZIKV infectivity (micro-neutralization). This assay was performed as described previously by Lopez-Chamaco et al., 2018. Briefly, at 72 h post-treatment (h.p.i.) the infected cell medium was removed, and cells were washed with PBS and then lysed with lysis buffer (20 mM Tris-HCl (pH 7.4), 20 mM iodoacetamide, 150 mM NaCl, 1 mM EDTA, and 0.5% Triton X-100) containing protease inhibitors. The lysis solution was transferred to an Immunolon^®^ plate pre-coated with 3 µg/mL of purified pan-flavivirus MAb D1-4G2-4-15 (ATCC^®^ HB112TM, herein referred to as 4G2) in PBS and incubated at room temperature for 1 h. After washing the plate with PBST solution, 100 µL of anti-ZIKV E polyclonal R34 IgG rabbit serum (dilution 1:5000) was added and incubated for 1 h. The plate was rinsed with PBST and incubated for 1 h with secondary HRP conjugate anti-rabbit IgG 7090 (Abcam, Ab 7090). After washing with PBST, 100 µL of TMB solution (Life Technologies) was added and the reaction stopped at 30 min with 50 µL of 0.5 M H_2_SO_4_ solution added and read at 450 nm using a plate reader spectrophotometer (ELISA) [55]. 

Entry step assessment of GA-Hecate and GA-metabolites on Zika Virus. Virucidal assay. ZIKV strain PE243 (MOI 0.1) was pre-incubated for 1 h with different concentrations of peptides before infecting the Vero cells for 1 h (7 × 10^3^ Vero cells/well). The solution was then removed, the cells were washed with PBS, a fresh medium containing 2% FBS was added, and the cells were incubated for 72 h at 37 °C. The levels of virus replication were determined using micro-neutralization as described above. Protective assay. Vero cells (7 × 10^3^ cells/well) were incubated for 1 h with peptides, following which they were washed three times with PBS and then infected for 1 h with ZIKV (MOI 0.1). The cells were lysed at 72 hpi and the lysates used to perform a micro-neutralization assay as described above. Water was used as a negative control. 

Immunofluorescence assay. A solution of 5 × 10^3^ PFU of ZIKV strain MP1751 was incubated for 1 h with 40 µM of peptides before infecting 5 × 10^4^ Vero cells (0.1 MOI) for 1 h in a 24-well plate, and the cells were then washed with PBS and refed with fresh medium containing 2% FBS and incubated at 37 °C for the indicated time. After incubation, the supernatants were frozen at −80 °C and the cells were fixed with ice-cold methanol for 2 h at −20 °C. After washing with PBST, fixed cells were incubated with 4G2 antibody at 3 µg/mL for 1 h, washed, and then stained with donkey anti-rabbit Alexa Fluor-488 fluorescent dye antibodies (Dilution 1:500). Images were obtained using fluorescence and confocal microscopes.

Focus-forming assay. The 48 h and 72 hpi supernatants derived from cells used in the immunofluorescence above were used to infect naïve Vero cells in a 96-well plate (7 × 10^3^ cells/well) using 10-fold dilutions (starting 1:10) in 8 replicates for 12 concentrations. After 1 h, the virus solution was removed, and cells were incubated for 48 h at 37 °C in 100 µL of 2% FBS medium. Cells were washed with PBS and fixed with ice-cold methanol for 2 h at −20 °C. After washing with PBST, fixed cells were incubated with 4G2 antibody at 3 µg/mL for 1 h, washed, and then stained with donkey anti-rabbit Alexa Fluor^®^-488 fluorescent dye antibodies (dilution 1:500). Individual foci were counted using a fluorescence microscope and focus-forming unities/mL (FFU/mL) were obtained using the Spearman and Kerber calculation method [52]. 

Virus assembly/release. Vero cells (7 × 10^3^ cell/well) were infected for 1 h with ZIKV strain PE243 or ZIKV strain MP1751 (MOI 0.1) and then treated with 40 µM and 20 µM of peptide solutions in 2% FBS culture medium. The supernatant was removed at 48 hpi, and cells were washed with PBS and fed with fresh medium containing 2% FBS (DMEM). After 24 h, supernatants were collected and 100 µL was used to infect Vero cells in a 96-well plate for 1 h at 37 °C. The cells were then washed with PBS and fed with fresh medium containing 2% FBS. After 72 hpi, cells were lysed, and the lysates were used to determine the levels of virus replication in a microneutralization assay as described above. 

GA-Hecate and GA-metabolite cell toxicity during constant treatment. Vero cells were cultured in the DMEM medium (described above) containing 20 µM of peptides and maintained at 37 °C in a humidified 5% CO_2_ incubator for 3 and 7 days. Prior to seeding in a 96 well-plate and incubation with medium lacking peptides for 24 h, the cells were treated with the proteolytic enzyme, trypsin. Peptide media were removed, and WST-1 at a dilution of 1:10 was added to the wells and incubated for 30 min. Absorbance was measured at 450 nm, using a plate reader spectrophotometer. During the treatment of 7 days, cells were treated with trypsin after 3 days, keeping 1 mL of cells in the flask. Additionally, supplemented medium with compounds was renewed, maintaining the treatment for 4 more days. Cell images were obtained using a confocal microscope and the number of cells/mL was automatically measured using the automated cell counter Scepter^TM^ (Millipore). Treated cells remained for an additional 3 and 7 days without compounds to evaluate cell recovery. 

Statistical analysis. Statistical analyses were performed using one-way ANOVA with Tukey’s post hoc test. Viability and antiviral assays were performed in triplicate in three independent assays. All statistical tests were performed using GraphPad Prism 5.0 software (GraphPad Software, San Diego, CA, USA).

## 4. Discussion

Peptides have several advantages over small molecules [38,39,40,41]. They are more specific for targets, interact with proteins and cellular components, are biodegradable, and their composition can be designed, synthesized, and modified [39,56]. However, due to their low in vivo stability and rapid renal clearance, many promising peptides have failed to progress into the clinic [57,58]. Proteases can cleave peptide bonds, increasing/decreasing their biological activity, toxicity, half-life, depuration rate, organ distribution, excretion, etc. [53,59]. The evaluation of peptide serum stability constitutes a powerful and important screening strategy for drug discovery [53]. We identified six proteolytic products (N-terminal and C-terminal metabolites) of our GA-Hecate peptide following its incubation with human serum. The N-terminal metabolites were synthesized and conjugated with gallic acid (GA) [33]. The synthesized compounds vary in length, molecular weight, charge, and water solubility. The GA-metabolites present a carboxyl group (-COOH) at the C-terminal region while GA-Hecate presents an amide group (-CONH_2_) at the same position. Phenylalanine (F), alanine (A), leucine (L), and lysine (K) are common amino acids in all sequences and the percentages of these amino acids are variable for each fragment: GA-Hecate (F: 4.2%; A: 25%; L: 29%; K: 37.5%), GA-metabolite 5 (F: 6.2%; A: 25%; L: 31.2%; K: 31.2%), GA-metabolite 6 (F: 11.1%; A: 33.3%; L: 33.3%; K: 11.1%); and GA-metabolite 7 (F: 7.7%; A: 31%; L: 31%; K: 23%). The basic amino acid lysine on the side chain and N-terminal group determines the positive net charge of the fragments, while the C-terminal region is negatively charged (in that case only for GA-metabolites) at physiologic pH (~7.2). GA-Hecate is the most positive bioconjugate (+9), followed by GA-metabolite 5 (+4), GA-metabolite 7 (+2), and GA-metabolite 6 (0). All net positively charged compounds were soluble in water. GA-metabolite 6 had poor water solubility and therefore was not used for biological tests. 

Both GA-Hecate and GA-metabolite 5 inhibited ZIKV strain PE243 infection post-entry, albeit at different potency. This difference in relative potency can be explained by differences in net positive charge carried by each structure. GA-Hecate, the most positively charged compound (+9), decreased the replication rate for ZIKV strain PE243 dose-dependently by 20% to 60%, while GA-metabolite 5 (net charge + 4) decreased virus replication by 40% but only at 40 µM. GA-Hecate inhibited ZIKV strain MP1751 replication post-entry in a similar fashion; however, GA-metabolite 5 was ineffective against this strain in the post-entry setting. These results could be associated with peptide–RNA interactions, where nucleic acids present a net negative charge and the bioconjugate a net positive charge. Charge reduction has been implicated in lesser electrostatic interaction with viral RNA, which may in turn lead to lesser inhibition of virus replication. 

Flavivirus virions contain a positive-strand RNA genome that is encapsulated by the viral nucleocapsid protein. The nucleocapsid is enclosed within a host-derived lipid membrane with integrated viral glycoproteins [32,60,61]. Compounds that can target virions are a developing strategy and have been used by health organizations as prophylactic methods to avoid virus infection and control epidemics [62]. Compounds such as nonoxynol-9, para-aminobenzoic acid, fullerene derivatives, and anthraquinones have been reported as virucidal compounds against the enveloped virus [63,64,65,66,67]. Mastoparan, an invertebrate host defence peptide that penetrates lipid bilayers, and its derivatives interact with viral lipid envelopes, and thereby reduce the infectivity of West Nile virus, avian influenza, and henipavirus [68]. Amphipathic peptides with high affinity to lipids could interact with the viral envelope [33,69,70] to destroy virus particles by forming a pore (toroidal pore or barrel stave) or dissolving the membrane (carpet-like) [71]. Additionally, peptides could interact with virus structural proteins and avoid infection. However, more studies are needed to obtain information about the mechanism of action of peptides on the virus particle. Our results show that GA-metabolite 5 is a virucidal agent decreasing the ZIKV replication rate at all tested concentrations, while GA-Hecate showed virucidal activity only at 40 µM. 

Flavivirus enter host cells by endocytosis mediated by virus interaction with cellular receptors and lipids from the host cell membrane [28]. Studies have indicated that glycosaminoglycans (GAGs) are a class of receptors that mediate ZIKV and other sources of flavivirus entry in cells [72]. GAGs are long, unbranched, sulphated polysaccharides that are linked to core proteins attached to the cellular surfaces of all tissues. Flavivirus binding to cells occurs via electrostatic interactions between the negatively charged clusters of GAG and the positively charged amino acid residues of the viral E glycoprotein [28,73]. Another class of receptors involved in flavivirus entry belongs to the TIM/TAM family of proteins [74]. They are cell surface glycoproteins that bind phosphatidylserine (PS) on the surface of apoptotic cells and serve as flavivirus entry factors [75]. Compounds that interact with possible viral receptors may block virus entry and thus have prophylactic potential. Here, we explored the possibility of GA-peptides having such protective potential in addition to displaying virucidal effects. Indeed, GA-Hecate exhibited both virucidal (~47%) and protective (43.9%) activities against ZIKV strain PE243 at 40 µM, a concentration at which its virucidal activity was influenced by the protective activity. Interestingly, however, GA-Hecate at 40 µM displayed virucidal (~47%) but not protective activity against ZIKV strain MP1751. GA-metabolite 5 on the other hand was virucidal against ZIKV strain PE243 at all tested concentrations (74.6% at 40 µM and 20 µM; 45.9% at 10 µM). GA-metabolite 5 at 40 µM was both protective (86.7%) and virucidal (98.5%) against MP1571; however, at 20 µM it exhibited high virucidal activity (98.5%) but did not affect virus entry, indicating that the nature of the antiviral activity was dependent on peptide concentration. Additionally, cells co-treated with GA-Hecate and GA-metabolite 5 showed a promising synergistic effect at 40 µM against ZIKV strain PE243 (inhibition of 92.4%). The differential protective efficacy exhibited by these compounds against the two strains could be explained by the possible affinity of ZIKV strain PE243 and ZIKV strain MP1751 for different cellular receptors. In addition, the improvement in the inhibition of virus replication rate upon co-treatment indicated that the GA-peptides interact with different cellular receptors that are important for virus infection. On the other hand, GA-Hecate and GA-metabolite 5 were not able to inhibit the assembly and secretion of ZIKV from infected cells.

The virus needs a favourable cellular environment to generate new infectious particles. Viability assays using reagents that can measure cell viability are advantageous and provide important information about cell death after some treatments, especially after treatment with compounds [75]. However, the most used protocol uses a monolayer of 24 h pre-grown cells to test compounds and does not give information about the growing process. Additional experiments should be performed to confirm whether the tested compounds affect cell metabolism, cell growth, and virus replication. Thus, Vero cells were incubated consecutively for 3 and 7 days with 20 µM of compounds. In addition, after treatment, cells were incubated for an additional 3 and 7 days without compounds to evaluate cell growth recovery. GA-Hecate decreased cell viability by 40% after 3 and 7 days of incubation and after 3- and 6-days post-incubation. Without compounds, cells were able to recover viability, showing 100% viability in both cases. In contrast, GA-metabolite 5 did not considerably affect cell viability, even after 7 days. These results suggest that GA-Hecate modifies the cell metabolism, which affects cell growth and, possibly, virus replication into the cytoplasm. Therefore, we cannot exclusively attribute the previously shown antiviral post-entry results of this compound to virus inhibition. However, in the case of GA-metabolite 5, the lack of a decrease in cell viability in this experiment suggests that its post-entry activity is exclusively attributed to virus replication inhibition. 

## 5. Conclusions

In conclusion, upon incubation with serum, the Hecate peptide was found to be processed into a series of metabolites, which were then identified. These metabolites were subsequently synthesized and conjugated with gallic acid (GA). GA-metabolite 5 (GA-FALALKALKKALKKL-*COOH*) was the most efficient and non-toxic in inhibiting the main replication steps of two ZIKV strains, thus showing promising antiviral activity. Finally, we proposed a mechanism of action for this new compound, highlighting its activity in the entry and post-entry process during ZIKV infection. GA-metabolite 5 is a new synthetic antiviral to be used against the ZIKA virus, showing high potential to avoid infection and inhibit virus multiplication.

## 6. Patents

The compound GA-metabolite 5 has a patent deposit in Brazil registered in the National Institute of Industrial Property (INPI), under the number BR1020210147377.

## Figures and Tables

**Figure 1 molecules-28-04884-f001:**
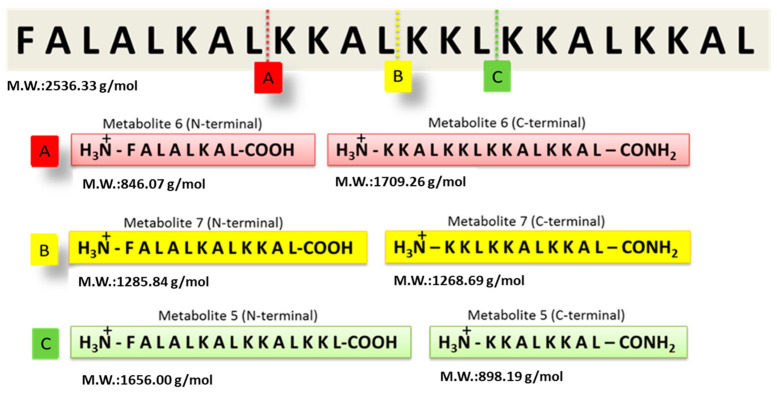
Identification of Hecate peptide fragments after incubation with human blood serum. Following incubation in freshly prepared human serum, the Hecate peptide was subjected to mass spectrometry and the proteolytic metabolites were identified as described in the text. The figure shows the cleavage sites of the Hecate peptide and the peptide sequence of the N- and C-terminal fragments (metabolites). (**A**) Metabolite 6, (**B**) Metabolite 7, and (**C**) Metabolite 5. M.W.: molecular weight. Water was used as the negative control replacing the volume of the blood serum solution.

**Figure 2 molecules-28-04884-f002:**
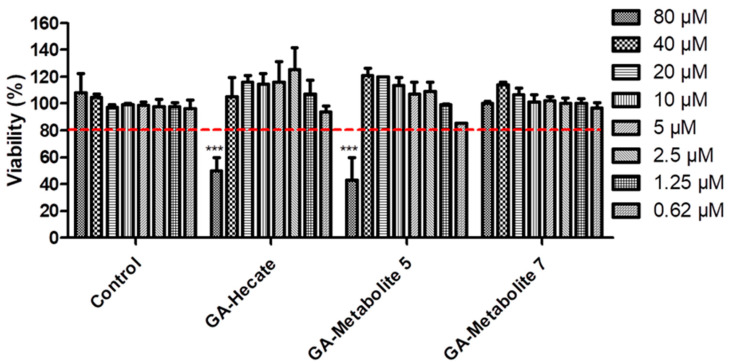
Compound effects on Vero cells. Vero cells were incubated with different concentrations of GA-peptides. At 72 h post treatment, cell viability was determined by WST viability assay and the values are plotted as those relative to the control. Bars represent triplicate samples of three independent assays. *** *p* < 0.001. Comparison control vs. concentration. Water, the GA-peptide vehicle, was used as a negative control. Red dotted lines represent the arbitrary threshold of 80% cell viability.

**Figure 3 molecules-28-04884-f003:**
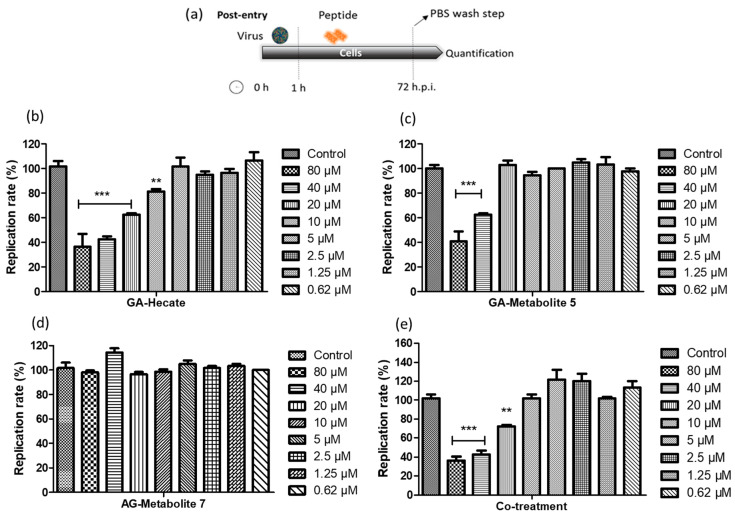
ZIKV strain PE243 replication percentage. (**a**) Schematic representation of the post-entry experiments. All dashed lines on the picture represent the PBS wash step. (**b**) GA-Hecate, (**c**) GA-metabolite 5, (**d**) GA-metabolite 7 and (**e**) co-treatment (GA-Hecate + GA-metabolite 5) after 72 hpi. Bars represent the standard deviation from triplicate samples of three independent assays. The ANOVA means control vs. concentration where *** *p* < 0.001; ** *p* < 0.01. Water was used as the negative control.

**Figure 4 molecules-28-04884-f004:**
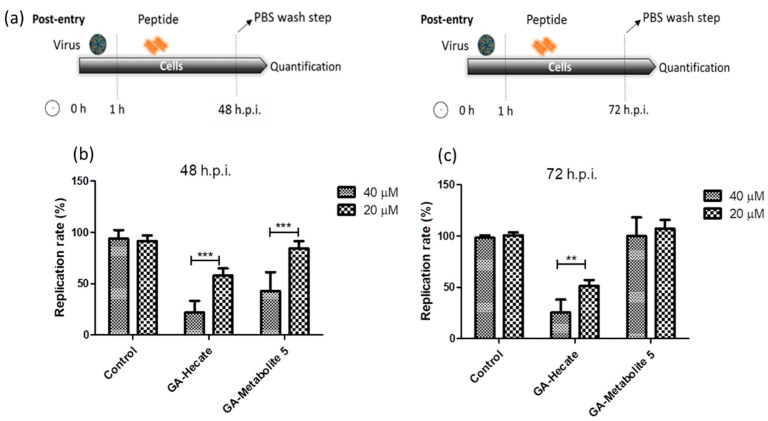
Effect of GA-Hecate and GA-metabolite 5 on ZIKV strain MP1751 replication. (**a**) Schematic representations of the post-entry experiments at 48 and 72 hpi, respectively. All dashed lines on the picture represent PBS wash steps. (**b**) A total of 48 hpi and (**c**) 72 hpi of treatment with 40 and 20 µM of GA-Hecate and GA-metabolite 5. Bars represent triplicate samples of three independent assays. *** *p* < 0.001; ** *p* < 0.01. Water was used as the negative control.

**Figure 5 molecules-28-04884-f005:**
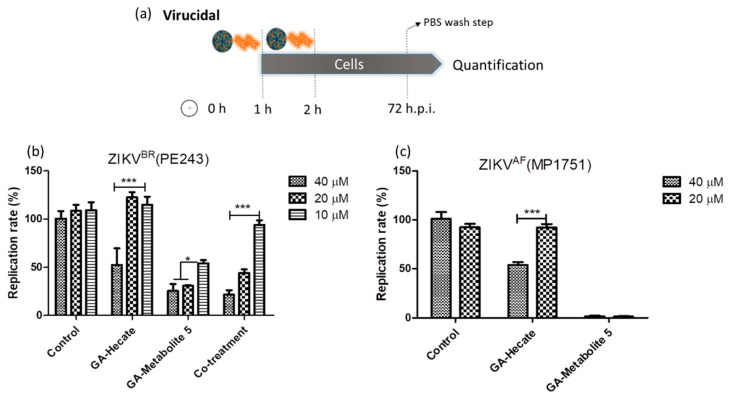
Virucidal effects on ZIKV strain PE243 and ZIKV strain MP1751 of GA-Hecate and GA-metabolite 5. (**a**) Schematic representation of the virucidal experiments. All dashed lines on the picture represent the PBS wash step. Replication rate at 72 hpi of treatment with 40, 20 and 10 µM of GA-Hecate and GA-metabolite 5 is shown in (**b**) for ZIKV strain PE243 and in (**c**) with 40 and 20 µM for ZIKV strain MP1751. Bars represent triplicate samples of three independent assays. *** *p* < 0.001. * *p* < 0.05. Water was used as the negative control. Co-treatment for ZIKV strain MP1751 was not shown once GA-metabolite 5 was almost completely inhibiting virus multiplication; the combination here is not appropriate.

**Figure 6 molecules-28-04884-f006:**
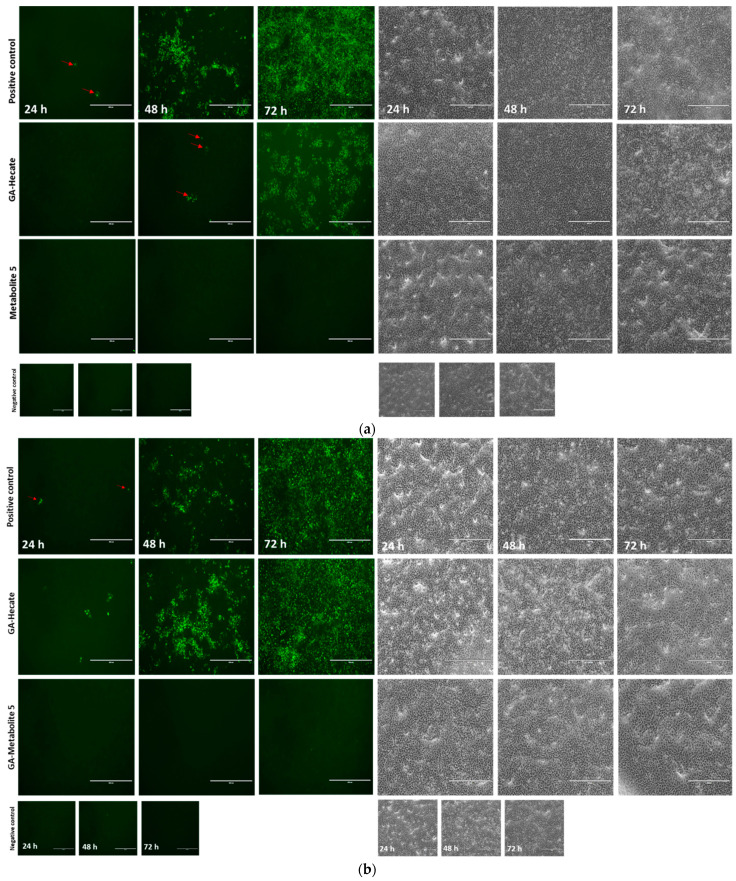
Fluorescence microscopy at 24, 48, and 72 h (hpi) of Vero cells infected with MP1571 after virucidal treatment. Vero cells were infected at an MOI of 0.1 with ZIKV strain MP1751, which had been pre-treated with (**a**) 40 µM or (**b**) 20 µM of GA-Hecate of GA-metabolite 5 for 1 h at 37 °C. Cells were washed with PBS and then incubated at 37^o^C for an additional 24, 48, and 72 h. The cells were then probed with anti-ZIKV E MAb 4G2, followed by anti-mouse IgG-Alexa Fluor-488, and subjected to fluorescent microscope analysis. Bars represent a scale of 400 µm. Arrows in red indicate the fluorescent focus. Water, the GA-peptide vehicle, was used as a negative control.

**Figure 7 molecules-28-04884-f007:**
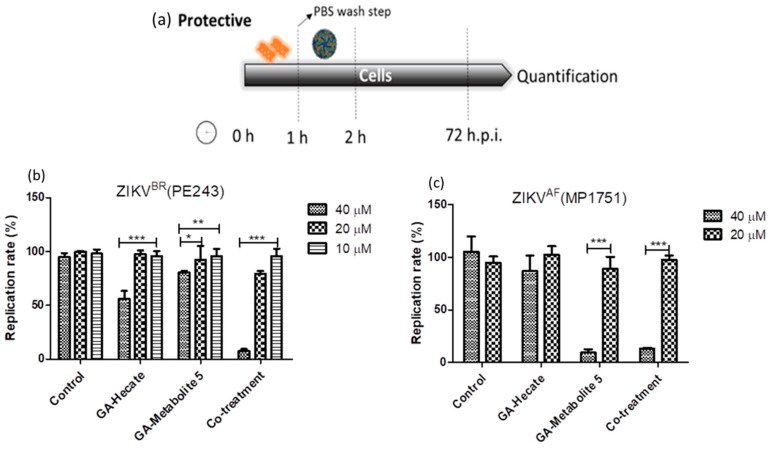
Protective effects on ZIKV strain PE243 and ZIKV strain MP1751 of GA-Hecate and GA-metabolite 5. (**a**) Schematic representation of the protective experiments. All dashed lines on the picture represent the PBS wash step. Replication rate at 72 hpi of treatment with 40, 20 and 10 µM of GA-Hecate and GA-metabolite 5 are shown in (**b**) for ZIKV strain PE243 and in (**c**) with 40 and 20 µM for ZIKV strain MP1751. Bars represent triplicate samples of three independent assays. *** *p* < 0.001; ** *p* < 0.01; * *p* < 0.05. Water was used as the negative control.

**Figure 8 molecules-28-04884-f008:**
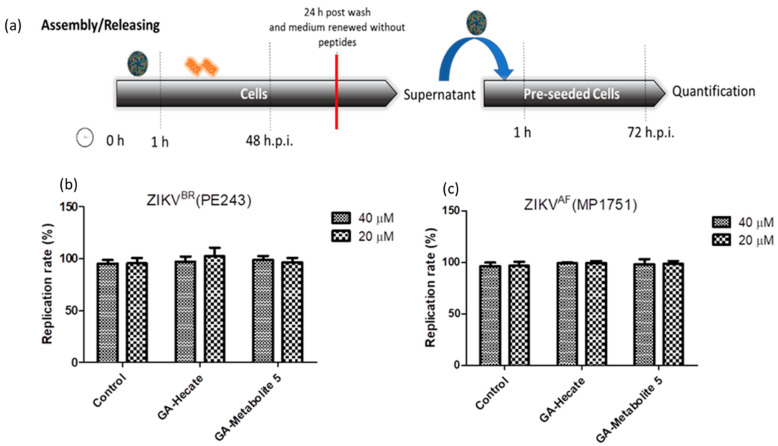
Assembly/release of ZIKV strain PE243 and ZIKV strain MP1751 after incubation and washing of the compounds. (**a**) Schematic representation of the assembly/release experiment. All dashed lines represent the PBS wash step. The red line represents the transfer of the supernatant to pre-seeded cells. (**b**) Replication rate of ZIKV strain PE243 and (**c**) ZIKV strain MP1751, respectively, after treatment with different concentrations of GA-Hecate and GA-metabolite. Bars represent triplicate samples of three independent assays. Water was used as the negative control.

**Figure 9 molecules-28-04884-f009:**
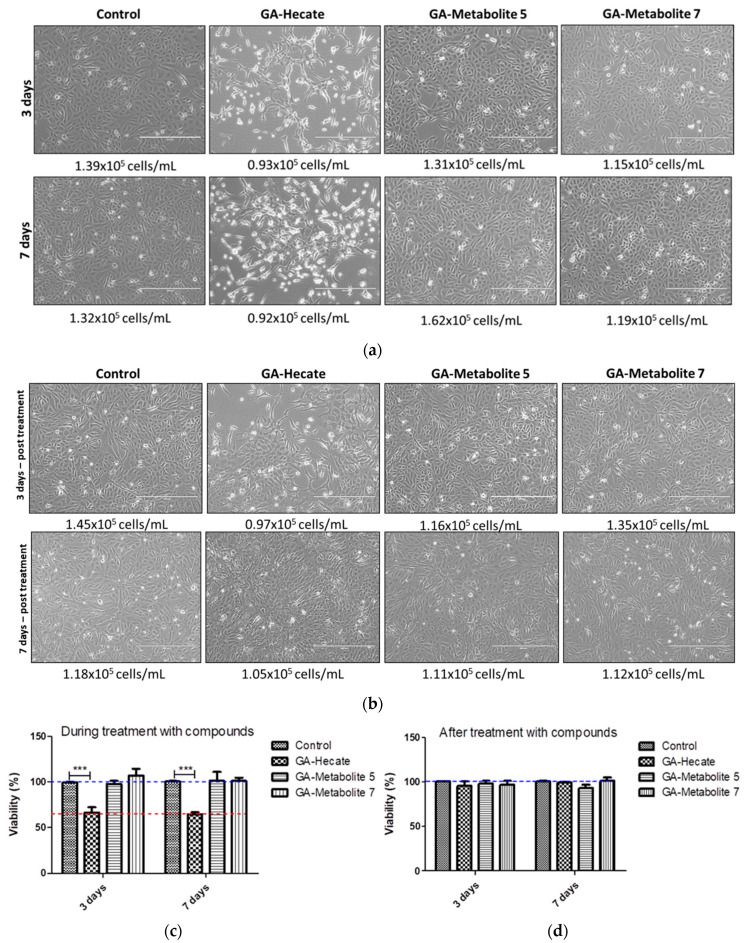
Effect of peptides on Vero. (**a**) Cells were treated with 20 µM each of the peptides and cell counts determined at 3 and 7 days (**a**) during or (**b**) after treatment, as described in the text. (**c**,**d**) depict the cell viability counts from (**a**,**b**), respectively, shown as plots. Bars represent triplicate samples of three independent assays. *** *p* < 0.001. Water was used as the negative control. Blue and red dotted lines represent 100% and 65% cell viability, respectively.

**Figure 10 molecules-28-04884-f010:**
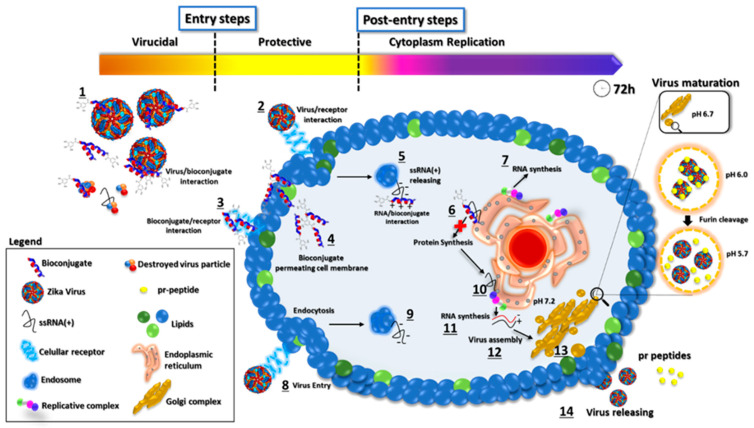
Proposed mode of action of GA-metabolite 5. The picture represents the viral replication process and the proposed mechanism of action of GA-metabolite 5 in terms of its effect on virus infection (entry steps) and replication (post-entry steps). The described compound can interact with virus particles and cellular receptors blocking virus entry. After infection, in the cytoplasm contents, the bioconjugate could interact with viral ssRNA(+), blocking proteins and, consequently, RNA synthesis. The process is shown in the figure: (1) Bioconjugate and virus particle interaction. (2) Virus binding on cellular receptors and endocytosis. (3) Bioconjugate interaction with the cellular receptor. (4) Bioconjugate permeating cell membrane. (5) ssRNA(+) release from endosome to cytoplasm and peptide/viral RNA interaction. (6) Polyprotein synthesis blocking by bioconjugate/RNA interaction. (7) RNA synthesis. (8) Virus entry and endocytosis. (9) ssRNA(+) release. (10) Polyprotein synthesis on the endoplasmic reticulum. (11) Viral RNA synthesis. (12) Virus assembly on Golgi complex. (13) Virus particle maturation. (14) Virus particle release.

**Table 1 molecules-28-04884-t001:** Peptides, their metabolites, and the identified m/z (mass/charge) relationship by mass spectrometry.

Peptide	Hecate	Metabolite 5		Metabolite 6		Metabolite 7	
Region		N-terminal	C-Terminal	N-terminal	C-terminal	N-terminal	C-terminal
Theoretical M.W. (g/mol)	2536.3	1656.0	898.2	846.1	1709.3	1285.8	1268.7
Experimental Results (*m*/*z*)	508.3 (+5) 635.0 (+4) 846.2 (+3)	1656.6 (+1)	301.2 (+3)	846.3 *		1285.5 (+1) 645.7 (+2)	1268.7 (+1)

* Not determined. Numbers in the brackets are the experimental charge (z) that corresponds to the experimental mass (m) obtained.

**Table 2 molecules-28-04884-t002:** Chemical properties of synthesized compounds. The table shows the amino acid sequences and the chemical properties of GA-peptides including molecular weight, charge, and solubility.

Name	Amino Acid Sequence	Molecular Weight (g/mol)	Net Charge ^1^ (pH 7.0)	Water ^2^ Solubility
GA-Hecate	GA-FALALKALKKALKKLKKALKKAL-*CONH_2_*	2688.4	+9	Soluble
GA-Metabolite 5	GA-FALALKALKKALKKL-*COOH*	1808.3	+4	Soluble
GA-Metabolite 6	GA-FALALKAL-*COOH*	998.2	0	Not soluble
GA-Metabolite 7	GA-FALALKALKKAL-*COOH*	1438.9	+2	Soluble

^1^ Calculated considering the charged groups from the amino acid sequence at pH 7.0. ^2^ The concentration of 1 mg/mL in pure water.

**Table 3 molecules-28-04884-t003:** Inhibition, cytotoxic concentrations, and uptake efficiency of GA-peptides.

Peptide	IC_50_ (µM) ^a^	CC_50_ (µM) ^b^	Uptake Efficiency ^c^
GA-Hecate	19.5	>70.0	Low
GA-Metabolite 5	>40.0	>70.0	High
GA-Metabolite 7	>80.0	>80.0	Low
Co-treatment	>40.0	>70.0	n.d.

^a^ IC_50_ (50% inhibition concentration for ZIKV strain PE243. ^b^ CC50 (50% cytotoxic concentration in VERO cells). EC_50_ and CC_50_ were calculated using a non-linear regression through a dose–response curve (log[peptide] versus response). ^c^ Predicted by MLCPP 2.0 online software [50].

**Table 4 molecules-28-04884-t004:** Comparison of the focus-forming unit (FFU) assays in Vero cells in GA-Hecate and GA-metabolite 5 after 48 and 72 hpi. The numbers represent the value of FFU/mL followed by the standard error of three independent assays.

	FFU/mL		
Concentration/Time	Control	GA-Hecate	GA-Metabolite 5
40 µM			
48 hpi	9.84 × 10^5^ ± 0.52	9.84 ± 0.23	-
72 hpi	7.38 × 10^6^ ± 0.08	3.11 × 10^4^ ± 0.6	7.38 ± 0.82
20 µM			
48 hpi	9.31 × 10^5^ ± 0.65	9.63 × 10^5^ ± 0.5	-
72 hpi	7.42 × 10^6^ ± 0.45	9.38 × 10^6^ ± 0.5	9.63 ± 0.52

## Data Availability

The data shown in this paper can be requested by e-mail to paulo.sanches@unesp.br.

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
