# Peer review of "Antiviral Evaluation of New Synthetic Bioconjugates Based on GA-Hecate: A New Class of Antivirals Targeting Different Steps of Zika Virus Replication"

_molecules, 2023, doi:10.3390/molecules28134884_

Round 1
Reviewer 1 Report
The article "Antiviral evaluation of new synthetic bioconjugates based on GA-Hecate: A new class of antivirals targeting different steps of Zika Virus replication" by Sanches et al, investigates the effect of GA-Hecate peptide and its metabolites on replication of ZIKV. The work is generally well written and the results well presented.
Some comments:
1) It would be nice to see the effect of free GA compared to GA-peptide conjugates. As understood, this is a continuation of previous paper where GA-Hacate was investigated. The comparsion of GA with the conjugates would suffice in the SI, as a supporting data for conclusions.
2) Does the net charge of the GA-peptide conjugate also affect the efficacy of the inhibition?
3)The results in Figure 9a, peptides treated with GA-Hecate and Figure 2. The figures reflect the cell viability after different treatment regimes. Did the cells show similar reduction also with lower GA-hecate concentrations?
4) The metabolite 5 sequence is a predicted cell-penetrating peptide with high internalisation efficacy (for example predicted with MLCPP 2.0), whereas Hecate and Metabolite 7 have lower internalisation efficacy. Would the authors also consider adding a short sentence about this? How it could affect the general efficacy, if indeed the Metabolite 5 is an efficient CPP.
Minor notes:
Figure 3. Disproportionate layout. Images distorted.
How were the peptide net charges calculated?
English Language is adequate
Author Response
Dear Reviewers,
We would like to express our gratitude for taking the time to review our manuscript titled " Antiviral evaluation of new synthetic bioconjugates based on GA-Hecate: A new class of antivirals targeting different steps of Zika Virus replication " submitted to Molecules. We appreciate your insightful comments and suggestions, which have undoubtedly contributed to the improvement of our work. In this response letter, we address each of your comments and provide a detailed account of the revisions we have made to the manuscript.
Reviewer 1
1) It would be nice to see the effect of free GA compared to GA-peptide conjugates. As understood, this is a continuation of previous paper where GA-Hacate was investigated. The comparsion of GA with the conjugates would suffice in the SI, as a supporting data for conclusions.
In our previous work describing the synthesis and toxicity of Hecate, Gallic Acid (GA) and GA-Hecate (https://doi.org/10.1007/s00726-015-1980-7) we have shown that GA in HaCat cells (non-tumor cells) presented higher toxicity compared with GA-Hecate at the same conditions. Additionally, in our work “GA-Hecate antiviral properties on HCV whole cycle represent a new antiviral class and open the door for the development of broad-spectrum antivirals” (DOI:10.1038/s41598-018-32176-w), we have shown that the GA-Hecate were the most non-. toxic and efficient compound against HCV using Huh-7.5 cells among another N-terminal Hecate derivatives, including (GA)2-Hecate. Due these results, we have not evaluated the free Gallic acid in this work, once the GA-Hecate is our hit compound used for comparison.
2) Does the net charge of the GA-peptide conjugate also affect the efficacy of the inhibition?
We believe that the Net-charge together with the structure of the compounds affect the efficacy of inhibition. In our previous work “GA-Hecate antiviral properties on HCV whole cycle represent a new antiviral class and open the door for the development of broad-spectrum antivirals” (DOI:10.1038/s41598-018-32176-w) we have shown that the most positive peptide (Lys-Hecate), the least positive peptide with the steric effect (GA-Hecate) and the Hecate general structure effects on dsRNA intercalation in the HCV replication. As the RNA molecules have Net-negative charges, we believe that the electrostatic interaction between the peptides (positive charged) and the RNA could be directed related to the increasing of the charges. Also, comparing the GA-metabolite 5 and the GA-metabolite 7 at the post-entry steps (described at this manuscript) we can see that GA-M5 (+4) was more efficient to inhibit ZIKV multiplication than GA-M7 (+2). As the GA-Hecate affected the cell growing at the constant treatment is hard to measure the role of the charge in the post-entry effects for this molecule.
3)The results in Figure 9a, peptides treated with GA-Hecate and Figure 2. The figures reflect the cell viability after different treatment regimes. Did the cells show similar reduction also with lower GA-hecate concentrations?
At the experiments to get the results showed in Figure 9a we only tested the concentration of 20 µM because was the lower concentration where we have seen post-entry inhibition of ZIKV multiplication in GA-Hecate. Also, at the same concentration the peptide has not shown toxicity in the viability assays. Additionally, we choose this concentration because it was the lower concentration that the GA-Hecate and GA-Metabolite 5 presented virus multiplication inhibition. We have not tested lower concentrations than 20 µM.
4) The metabolite 5 sequence is a predicted cell-penetrating peptide with high internalization efficacy (for example predicted with MLCPP 2.0), whereas Hecate and Metabolite 7 have lower internalization efficacy. Would the authors also consider adding a short sentence about this? How it could affect the general efficacy, if indeed the Metabolite 5 is an efficient CPP.
Thanks for the suggestion. We have added a short sentence about this at the table 3 and the text below of it.
Minor notes:
Figure 3. Disproportionate layout. Images distorted.
We changed the layout of the graphics.
How were the peptide net charges calculated?
The peptides net charges were calculated using the online software “Peptide property calculator” from Innovagen (PepCalc.com - Peptide calculator) ; and also manually considering the pKa of the side chain of the amino acids, especially the Lysine.

Reviewer 2 Report
In this study, the authors conducted a comprehensive evaluation of the serum stabilities, antiviral activities, virucidal effects, and protective effects of a gallic acid-conjugated peptide called GA-Hecate peptide, as well as its metabolites. Notably, among the metabolites examined, GA-Metabolite 5 demonstrated the highest efficacy in terms of antiviral activity against two strains of ZIKA virus, while maintaining a non-toxic profile on normal cell lines. Overall, these findings highlight the potential of GA-Metabolite 5 as a promising candidate for further development as an antiviral agent.
Major comments
1. (page 5, line 1) In the serum stability, the Hecate peptide was incubated in the human blood serum at 37℃ for 20 min. However, different experimental conditions, such as blood, plasma, or serum, contain various proteases, resulting in the generation of different metabolites, which may possess varying degrees of activity. How did the authors determine the experimental conditions to generate metabolites 5, 6, and 7? The following literature can be referenced for studies on proteolytic degradation of peptide-based drugs: doi: 10.1371/journal.pone.0178943
2. (Section 3.4.2, figure 5B) The GA-metabolite 5 exhibited approximately 75% inhibition at 20 µM and 45.9% inhibition at 10 µM. However, when GA-metabolite 5 was co-treated with GA-Hecate peptide, the inhibitory effect decreased. The authors should discuss the combined effect of GA-metabolite 5 and GA-Hecate peptide in their analysis.
3. (Figure 5C) According to the findings presented in Figure 5B, it is evident that the presence of GA-Hecate peptide has an impact on the activity of GA-metabolite 5. Consequently, the inclusion of co-treatment with GA-Hecate peptide is indispensable when studying the ZIKV MP1751 strain, and it is necessary to incorporate additional experimental conditions at 10 µM for meaningful comparison with the Figure 5B.
4. (Figure 7b) What are the possible reasons for the enhanced inhibition activity of GA-metabolite 5 when co-treated with GA-Hecate peptide, compared to GA-metabolite 5 alone at 40 µM?

Author Response
Dear Reviewers,
We would like to express our gratitude for taking the time to review our manuscript titled " Antiviral evaluation of new synthetic bioconjugates based on GA-Hecate: A new class of antivirals targeting different steps of Zika Virus replication " submitted to Molecules. We appreciate your insightful comments and suggestions, which have undoubtedly contributed to the improvement of our work. In this response letter, we address each of your comments and provide a detailed account of the revisions we have made to the manuscript.
- (page 5, line 1) In the serum stability, the Hecate peptide was incubated in the human blood serum at 37℃ for 20 min. However, different experimental conditions, such as blood, plasma, or serum, contain various proteases, resulting in the generation of different metabolites, which may possess varying degrees of activity. How did the authors determine the experimental conditions to generate metabolites 5, 6, and 7? The following literature can be referenced for studies on proteolytic degradation of peptide-based drugs: doi: 10.1371/journal.pone.0178943.
The serum stability protocol is well established in our research group. We use the freshly prepared serum and plasma together in a 25% solution in PBS to better identify the metabolites that could be generated in an organism after inoculation or absorption. Different times of incubation were evaluated, but we just used the 20 min, where the visualization of the parameter m/z was clearer. To do the experiments we used the refence Hilpert, K.; Fjell, C.D.; Cherkasov, A. Peptide-Based Drug Design; 2008; Vol. 494; ISBN 978-1-58829-990-1. We have now cited the proposed reference as well (ref 48).
- (Section 3.4.2, figure 5B) The GA-metabolite 5 exhibited approximately 75% inhibition at 20 µM and 45.9% inhibition at 10 µM. However, when GA-metabolite 5 was co-treated with GA-Hecate peptide, the inhibitory effect decreased. The authors should discuss the combined effect of GA-metabolite 5 and GA-Hecate peptide in their analysis.
We have done the discussion in the text.
The co-treatment has shown a decreasing of inhibition compared with GA-Metabolite 5 at 20 and 10 µM. As the GA-Hecate has not shown virucidal activity at these concentrations, the molecule could be avoiding the GA-Metabolite 5 to interact with virus particle and blocking the virucidal effects.
- (Figure 5C) According to the findings presented in Figure 5B, it is evident that the presence of GA-Hecate peptide has an impact on the activity of GA-metabolite 5. Consequently, the inclusion of co-treatment with GA-Hecate peptide is indispensable when studying the ZIKV MP1751 strain, and it is necessary to incorporate additional experimental conditions at 10 µM for meaningful comparison with the Figure 5B.
We completely understand the purpose of the reviewer about the co-treatment results at 10 µM for ZIKVAF. However, we didn’t add the results because the GA-Metabolite 5 presented an inhibition higher than 95% at 40 and 20 µM and, as we are evaluating a possible treatment in vitro, why should we test a co-treatment if just one single molecule has an efficient result? In other cases, we performed the co-treatment because the inhibition could be improved, but not for the results of the fig. 5C.
- (Figure 7b) What are the possible reasons for the enhanced inhibition activity of GA-metabolite 5 when co-treated with GA-Hecate peptide, compared to GA-metabolite 5 alone at 40 µM?
Both peptides presented protective effects in ZIKVBR at 40 µM (GA-Hecate 43.9%, GA-Metabolite 5 16.8%) and the inhibition was expressively higher when the co-treatment was evaluated, especially at the same concentration (40 µM, 92.4%). the improvement in inhibition of virus replication rate upon co-treatment indicated that the GA-peptides could be interacting with different cellular receptors that are important for virus infection. Also, the peptides together could interact with different parts of the same receptors and better inhibit the virus interaction with the host cells compared with the peptides alone. However, there are many other possibilities, and we are now better investigating the protective effects of these peptides in ZIKV infection, using Vero and another cell lineages such as BHK-21, PC3 and Huh-7.0.

Round 2
Reviewer 2 Report
In response to the major comments, the authors have provided thorough explanations. Based on these revisions, I believe that this manuscript is now suitable for publication.